



# Glacier-level and gridded mass change in the rivers' sources in the eastern Tibetan Plateau (ETPR) from 1970s to 2000

Yu Zhu[1,2], Shiyin Liu[1,2*], Junfeng Wei[3], Kunpeng Wu[1,2], Tobias Bolch[4], Junli Xu[5], Wanqin Guo[6], Zongli Jiang[3], Fuming Xie[1,2], Ying Yi[1,2], Donghui Shangguan[6], Xiaojun Yao[7], Zhen Zhang[8]

1 Yunnan Key Laboratory of International Rivers and Transboundary Eco-Security, Yunnan University, Kunming 650500, China

2 Institute of International Rivers and Eco-security, Yunnan University, Kunming, Yunnan 650500, China

3 School of Resource Environment and Safety Engineering, Hunan University of Science and Technology, Xiangtan, China

4 Institute of Geodesy, Graz University of Technology, Steyrergasse 30, Graz, Austria

5 Department of Surveying and Mapping, Yancheng Teachers University, Yancheng, 224002, China

6 Northwest Institute of Eco-Environment and Resources, Chinese Academy of Sciences, Lanzhou 730000, China

7 College of Geography and Environmental Sciences, Northwest Normal University, Lanzhou 730070, China

8 School of Geomatics, Anhui University of Science and Technology, Huainan 232001, China

*Correspondence: Shiyin Liu (shiyin.liu@ynu.edu.cn)

**Abstract.** The highly glacierized eastern part of the Tibetan Plateau is the key source region for seven major rivers: Yangtze, Yellow, Lancang-Mekong, Nu-Salween, Irrawaddy, Ganges, and Brahmaputra rivers. These rivers are vital freshwater resources for more than one billion people downstream for their daily life, irrigation, industrial use, and hydropower. However, the glaciers have been receding during the last decades and are projected to further decline which will profoundly impact the water availability of these larger river systems. Although few studies have investigated glacier mass changes in these river basins since the 1970s, they are site and temporal specific and limited by data availability. Hence, knowledge of glacier mass changes is especially lacking for years prior to 2000. We therefore applied digital elevation models (DEMs) derived from large scale topographic maps based on aerial photogrammetry from the 1970s and 1980s and compared them to the Shuttle Radar Topography Mission (SRTM) DEM to provide a complete picture of mass change of glaciers in the region. The mass changes are presented on individual glacier bases with a resolution of 30 m and are also aggregated into gridded formats at resolutions of 0.1° and 0.5°. Our database consists of 13117 glaciers with a total area of ~21695 km². The annual mean mass loss of glaciers is -0.30 ± 0.12 m w.e. in the whole region. This is larger than the previous site-specific findings, the surface thinning increases on average from west to east along the Himalayas-Hengduan mountains with the largest thinning in the Irrawaddy basin. Comparisons between the topographic map-based DEMs and DEMs generated based on Hexagon KH-9 metric camera data for parts in the Himalayas demonstrate that our dataset provides a robust estimation of glacier mass changes. However, the uncertainty is high in high altitudes due to the saturation of aerial photos over low contrast areas like snow surface on a steep terrain. The dataset is well suited for supporting more detailed climatical and hydrological analyses and is available at https://doi.org/10.11888/Cryos.tpdc.301236 (Liu et al., 2024).

Keywords: Glaciers; Mass change; Topo DEM




## 1 Introduction

Glaciers are important sources of fresh water impacting most of rivers and lakes in the High Mountain Asia (HMA) (Immerzeel et al., 2020; Immerzeel et al., 2013). They are highly sensitive to temperature and precipitation changes (e.g., Harrison, 2013; Wu et al., 2018). Driven by climate change, glaciers across High Mountain Asia (HMA) have exhibited accelerating mass loss in recent decades (Bolch et al., 2012; Bolch, 2019; Bhattacharya et al., 2021; Hugonnet et al., 2021; Shean et al., 2020). However, the glacier mass loss is not uniform, with the most severe

reductions in the southern and southeastern Tibetan Plateau, contrasting with balanced mass budgets observed in the Karakorum and Western Kunlun Shan (Kääb et al., 2015; Shean et al., 2020).

The Yangtze, Yellow, Lancang-mekong, Salween (Nu Chiang), Irrawaddy, Ganges, and Brahmaputra rivers originate from the eastern and southern Tibetan Plateau and the Himalaya collectively referred to as the extended Eastern Tibetan Plateau Region (ETPR) for brevity throughout this paper. There rivers provide a significant

contribution to the water supply of millions of people (Table 1 & Fig. 1) (Immerzeel et al., 2010). Data from the Randolph Glacier Inventory (RGI 6.0) reveals over 23,000 glaciers, encompassing a total area of $22782 \pm 683$ km$^2$, in the source area of these rivers (Pfeffer et al., 2014). Meltwater from these glaciers, in conjunction with precipitation, sustains river flow. Glaciers in ETPR are characterized by a high sensitivity to temperature fluctuations. This sensitivity is particularly pronounced when ice reaches the pressure-melting point, leading to

accelerated ablation rates (Shi et al., 1988). Glacier retreat across the region over the past three decades is well-documented (e.g., Zhang et al., 2012; Xu et al., 2018; Liu et al., 2019; Wu et al., 2020), with substantial and accelerating reductions in area, length, and thickness. This trend is likely driven by rising temperatures and a complex interplay of factors influencing ablation rates (Table 1). These factors include variations in weather patterns (Table 1), diverse topography (Hewitt, 2011), and the distribution of surface and surrounding features (e.g., debris

cover, ponds, crevasses, proglacial lakes) (Scherler et al., 2011; Watson et al., 2018; King et al., 2019). These combined factors pose a significant challenge to accurately evaluating and understanding glacier meltwater dynamics.

The elevation difference derived from the comparison of Digital Elevation Models (DEMs) is an effective approach to assess glacier mass balance (e.g., Bamber and Rivera, 2007). Recent researches based on remote sensing

data have enabled substantial advances in quantifying glacier mass change after 2000. Some studies utilizing the declassified spy satellite data and topographic maps have been able to extend the analysis back to the 1970s for specific regions in HMA, which provides insight into glacier mass balance prior to 2000 (Maurer et al., 2019; Ye et al., 2015). However, inconsistencies arising from data source variations hinder robust comparisons of glacier mass changes across the region (Shean et al., 2020; Hugonnet et al., 2021; Zhao et al., 2022). Furthermore, the limited

availability of pre-2000 data for many glaciers restricts comprehensive assessments of long-term mass changes and their sensitivity to climate change. This scarcity is particularly pronounced for the accumulation zones, where data voids within pre-2000 DEMs (e.g., KH-9 Hexagon) introduce uncertainties that compromise the accuracy of glacier contribution estimates to downstream water resources and sea level rise (Bhattacharya et al., 2021; Zhou et al., 2018). Additionally, there is a scarcity of gridded mass balance products derived from the aggregation of glacier-

level mass balance data at specific grid resolutions (e.g., 0.5°) that could serve energy balance simulation and hydrological simulation in a large glacierized basin in the ETPR before 2000. We therefore aim to (1) generate a set of glacier-level and gridded mass balance dataset from 1970s to 2000 based on the historical topographic maps (hereafter Topo maps) and the SRTM DEM; (2) provide a comprehensive view on the mass balance across the source area of the ETPR.


**Table 1**. Basic information of the southwestern river basins. Glacier areas were obtained from RGI 6.0. Precipitation and air temperature were sourced from Climatic Research Unit (CRU) (average from 1970 - 2000). Population for the year 2000 were acquired from the GPWv4 dataset. Irrigated areas for the year 2000 are reproduced from Siebert et al. (2015). Upstream is defined as the region with elevation higher than 2000 m.

|  | Ganges | Brahmaputra | Yangtze | Yellow | Mekong | Salween | Irrawaddy |
|---|---|---|---|---|---|---|---|
| Total area (km$^2$) | 952500 | 629551 | 1920292 | 963345 | 773231 | 265761 | 375475 |
| Upstream area (km$^2$) | 137046 | 374032 | 628021 | 353312 | 115936 | 126639 | 29373 |
| Glacier area (km$^2$) | 8019.78 | 10411.96 | 2109.56 | 212.52 | 214.66 | 1751.92 | 61.71 |
| Average temperature (℃) | 21.38 | 10.67 | 11.50 | 5.88 | 21.19 | 11.47 | 21.37 |
| Annual precipitation (mm) | 1063 | 1560 | 983 | 367 | 1453 | 986 | 1775 |
| Total Population (millions) | 391.326 | 118.716 | 465.404 | 119.516 | 48.960 | 8.155 | 27.845 |
| Irrigated area (km$^2$) | 275867 | 42363 | 168500 | 59239 | 24313 | 2043 | 11829 |

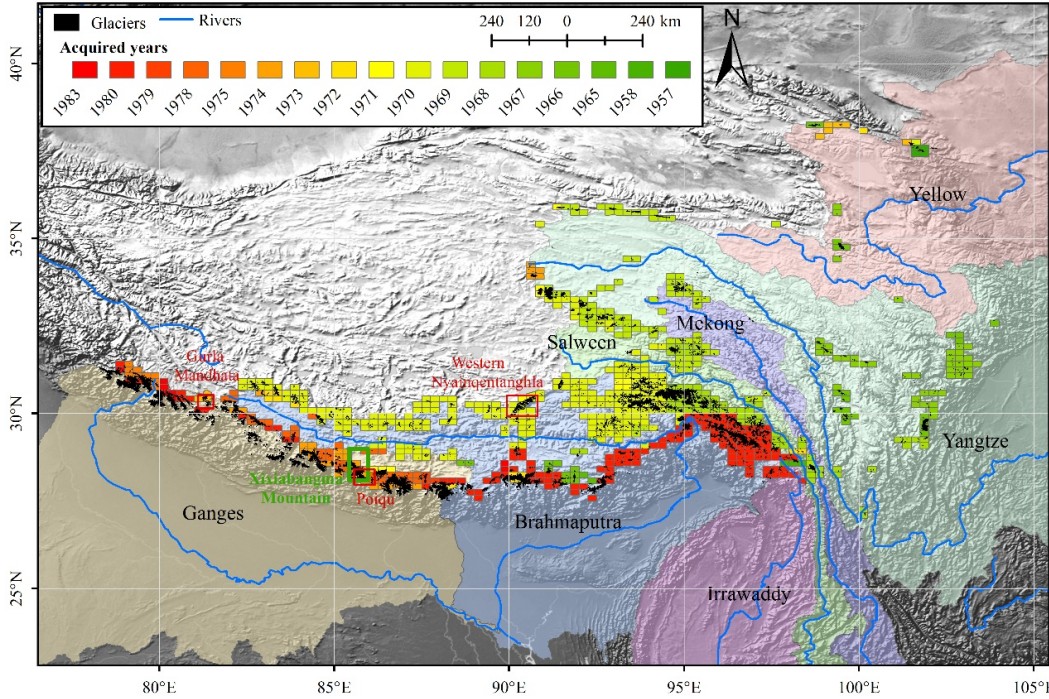

**Figure 1**. Glaciers in the ETPR and coverage of the Topographic maps. The maps were generated from aerial photos acquired during 1957−1983. The red and green edge rectangles denote specific regions used for product comparison between our dataset and others.

## 2 Data and methods

### 2.1 Topographic maps

We employed a total of 718 historical topographic maps including 142 at a scale of 1:50,000 and 576 at a scale of 1:100,000 compiled from aerial photos taken from 1957 to 1983 by the Chinese Military Geodetic Service (Fig. 1). These maps are referenced to 1954 Beijing Geodetic Coordinate System, which uses a datum level based on the





mean sea level of Yellow Sea observed at Qingdao Tidal Observatory in 1956. The vertical accuracy of the
topographic maps meets the National Photogrametrical Standard of China (NPSC, GB/T 12343.1-2008).
Specifically, the vertical error is less than 5 m on slopes less than 6° and ranges from 5 to8 m on slopes between
6°−25°. By using a Helmert transformation (also known as a seven-parameter transformation method), all scanned
maps were rectified based on gridded points of the meridians and wefts. The geometry accuracy is controlled by
over 600 ground control points measured by the State Bureau of Surveying and Mapping during 1950 to 1990.
Subsequently, the contours and other layers were manually digitized on-screen with the 1954 Beijing coordinate
system. These digital maps were later transformed into the coordinate system of SRTM, namely the World Geodetic
System 1984 (WGS84) / Earth Gravity Model 1996 (EGM96). To generate DEMs, the digital contours were first
used to build Triangulated Irregular Networks (TINs), and then the TINs were transformed to gridded elevation
products with the natural neighbour smoothing and interpolation method. The Topo DEMs were finally resampled
to a 30 m resolution to match the resolution of SRTM DEM. The uncertainties of Topo DEMs might originate from
operator proficiency during map generation, limitations in extracting elevation data from areas with low-contrast
features such as snow cover or shadows, and horizontal positioning errors. We will discuss these uncertainties in
Section 4.2.

Based on the historical topographic maps, the first glacier inventory of China (CGI1) was compiled by more
than 50 experienced technicians to display the status of glacier before 2000 (Shi, 2008). Limited by less-
sophisticated techniques and the scarcity of high-resolution satellite imagery, the first version has a margin of error
in glacier area exceeding 10%. To improve the accuracy of CGI1, the work group of the second glacier inventory
of China (CGI2) evaluated and refined the CGI1 glacier outlines using the advanced technology employed for CGI2.
The revised CGI1 was used for processing the elevation difference in this study.


### 2.2 SRTM

The SRTM mission provides C- and X-band topographic data in February 2000 with a 1 and 3 arc second special
resolution (Farr et al., 2007). The previous studies have pointed out that the accuracy of SRTM DEM (C-band)
presented a high level (-5.61 ± 15.68 m) for high relief (Carabajal and Harding, 2006). In glacierized regions, the
elevation bias can be up to 10 m at high altitudes (Berthier et al., 2006). In this study, we use SRTM DEM (no void
filled version) with a resolution of 1 arc second (~30 m) refer to the glacier surface in the year of 1999 suggested
by previous studies (e.g., Gardelle et al., 2013; Mcnabb et al., 2019). In addition, the limited X-band based DEM
data (Hoffmann and Walter, 2006) has been employed to quantify the underestimation of elevations in C-based
DEM caused by penetration of microwaves. All datasets are downloaded from http://earthexplorer.usgs.gov/.


### 2.3 KH-9 data

The KH-9 images, declassified in 2002, were acquired by a cold war–era spy satellites with a flight height of
approximately 170 km from 1971 to 1986. In this study, we acquired the KH-9 images covering Xixiabangma
Mountain. To generate a DEM, reseau grids overlaid on the KH-9 images were used to reconstruct the original
geometry at first (Pieczonka et al., 2013). Subsequently, surrounding pixels filled the grid areas, followed by image
enhancement techniques including histogram equalization and Wallis filtering. Manually identification of GCPs is
a conventional method to estimate unknown camera position and orientation, typically for a few selected image
pairs (Pieczonka et al., 2013). In this study, 26 evenly distributed GCPs were selected from the reference images
(Landsat ETM+ images for horizonal orientation and SRTM for vertical orientation) to correct the external
orientation of the KH-9 images. Several tie points in glacier regions were added manually to improve the matching
accuracy. By considering the focal length (12 inches for 1973-1980), flight height, and the film size (~ 9×18 inches),
the DEM were finally generated. In some case (e.g., Bhattacharya et al., 2021), a physical model was used to



compensate the unclassified (or unknown) imaging parameters (e.g., lens distortion). In addition, we obtained elevation difference products from KH-9 and SRTM published by Bhattacharya et al. (2021). These datasets were 150 used to evaluated the performance of elevation difference derived from the Topo DEMs.

**2.4 ICESat-2 data**

The ICESat-2, launched in 2018, is the second spaceborne laser altimetry mission which aims to estimate the ice sheet mass change, lake volume, and global vegetation heights (Markus et al., 2017). Similar to ICESat-1, ICESat-155 2 acquires the distance between the senor and Earth's surface by recording the travel time of the laser pulses, which are sent out by the Advanced Topographic Laser Altimetry System (ATLAS)— a photon-counting 532 nm (green light) lidar operating at 10 kHz. The system split the laser pulse into six beams forming 3 pairs (a weak and a strong beam separated by 90 m, included in one pair) to make an accurate measurement of surface height. It takes 91 days to complete a full orbital cycle (1,387 unique reference ground tracks) (Smith et al., 2019). ICESat-2 boasts denser 160 ground footprints compared to ICESat-1 due to its smaller footprint diameter (~17 m) and reduced along-track offset (~0.7 m) for each laser beam. The photon geolocation information is aggregated at 40 m along-track length scales.

**2.4 Coregistration and potential systematic errors correction for derived elevation difference**

**2.4.1 Coregistration between DEMs**

There are both horizontal and vertical offsets in the co-registered two DEMs due to dataset sources and the method used in production of DEMs (Paul, 2008). These shifts can be corrected by using an iterative co-registration method implemented by Nuth and Kääb (2011). In this method, the elevation difference in a static off-glacier surface between two DEMs (defined as $dh$) were correlated with the slope and aspect of the surface (Nuth and Kääb, 2011; Berthier et al., 2007). The relationship can be quantified by the Equation (1).

$$dh = a * cos(b - \varphi) * tan(\alpha) + \overline{dh} \quad (1)$$

Where $\varphi$ is aspect; $\alpha$ is slope; $\overline{dh}$ is the vertical offset of the registered DEM; $a$ and $b$ represent the horizonal offset and direction, respectively.

DEMs generated from aerial or satellite imagery are susceptible to rotational errors due to limitations in absolute accuracy. The coregistration method by Nuth and Kääb (2011) does not account for such errors. 175 Consequently, a complementary correction, such as deramping or iterative closest point (ICP) registration (Besl and Mckay, 1992) is required. In the current study, we defined the SRTM DEM as reference DEM. The off-glacier region was identified as static surface which was assumed to be stable during the study period (1970s - 2000). It should be noted that the processes for registration and correction of TOPO DEM were applied at a group-scale rather than map-scale. Every group including at least one TOPO DEM should cover all glaciers in an independent 180 mountain range. This group-scale processing strategy for generating elevation differences minimizes errors caused by glacier extents exceeding individual map sheet boundaries. There is a total of 261 groups in the ETPR. By conducting the co-registration, the resulting horizontal translation was applied to TOPO DEMs of each group and the median vertical bias was removed. Subsequently, we performed a 2nd order deramping for correction of possible rotations. Deramping method works by estimating and correcting for an N-degree polynomial over the entire 185 differences between a reference and the DEM to be aligned. Biases caused by different spatial resolutions between the DEMs could be adjusted by the relationship between elevation differences and maximum curvatures (Gardelle et al., 2013). The application of all corrections improved the accuracy of derived elevation differences in off-glacier regions, as evidenced by both the histogram statistics (Fig. S1) and the reduction in median and normalized median absolute deviation (NMAD) shown in Table S1, compared to the values observed before applying corrections.

The corrected elevation differences with values exceeding ±150 m were assumed to be obvious outliers and removed (Bhattacharya et al., 2021; Pieczonka and Bolch, 2015). There were still some unrealistic phenomena, such



as large surface thinning in accumulation zones, which are typical for Hexagon KH-9 based elevation changes (Zhou et al., 2018). These outliers were attributed to inaccurate matching of DEMs and can be removed by using an elevation dependent sigmoid function (Equation 2) (Zhou et al., 2018; Pieczonka and Bolch, 2015).


$$\Delta dh_{max} = \left(\frac{h_{max} - h_{gla}}{h_{max} - h_{min}}\right)^2 * A \quad (2)$$

Where, $\Delta dh_{max}$ represents the maximum elevation change, $h_{max}$, $h_{min}$, and $h_{gla}$ are maximum, minimum, and pixel-level elevation respectively, and $A$ is the maximum elevation change occurring within 150 m of each glacier terminus. Following outlier removal, normalized regional hypsometric interpolation was employed to fill remaining data gaps. In our data products, we have preserved the elevation changes without applying interpolation.


**2.4.2 Penetration depth estimation and correction of SRTM DEM**

DEMs derived from SRTM C-band contain biases due to the penetration of microwaves on dry snow, firn, and ice (Rignot et al., 2001). Since the narrow swath widths and limited available data (See Fig. S2), it is hardly to use X-band based DEM (e.g., Hoffmann and Walter, 2006; Dehecq et al., 2016) to calculate elevation difference. We

therefore used the X-band DEM to evaluate the penetration depth of C-band DEM ($C_{penetration}$). The penetration depth was estimated in the glacierized areas by comparing the SRTM C-band and X-band DEM data. To extend our results to the entire study area despite incomplete data coverage, we employed a limited-data fitting approach to establish a relationship between penetration depth and altitude. This relationship was then applied to all data. The results revealed an average penetration depth ranging from 3.8 to 6.2 m across the elevation range (Fig. 2 & Fig.

S3). While penetration depth increases with longer radar wavelength (Curlander and Mcdonough, 1991), it doesn't imply that X-band DEM presents no penetration effects. While some studies suggest this penetration depth is often negligible for estimating elevation changes over a decade, in the Qinghai-Tibet Plateau, it can exceed 2 m (Li et al., 2021), potentially influencing elevation differences.

The calculated penetration depth above represents the difference between C-band and X-band radar. To account for

X-band penetration, the study employed results from Zhou et al. (2022), who calculated X-band penetration in the subregion of ETPR using TanDEM-X and Pléiades DEMs with an acquisition time difference of only 4 days. Their results were used to fit a relationship between penetration depth ($X_{penetration}$) and altitude ($ele$), as shown in the following equation:

$$X_{penetration} = ele * 0.0068 - 34.3368 \quad (3)$$

This relationship was applied for X-band penetration correction. Ultimately, the C-band DEM-based penetration depth used in the study is the sum of the calculated C-X band difference ($CX_{diff}$) and the X-band penetration.

**2.5 Uncertainty estimation**

**2.5.1 Uncertainties in elevation difference estimation**

DEMs are inherently susceptible to large-scale instrument noise and variable vertical precision, resulting in complex error patterns. The accuracy and precision have been used to describe these errors. Precision pertains to random errors, whereas accuracy addresses systematic errors and, in instances, both systematic and random errors (Hugonnet et al., 2022). In many cases of studies on glacier elevation changes, the stable terrain was used for evaluating uncertainties in DEMs (Shean et al., 2020; Zhou and Duan, 2024). However, most of the previous studies

largely underestimated uncertainties in DEMs due to insufficient consideration of spatial correlation errors, particularly long-range spatial correlations. In this study, we leverage the comprehensive uncertainty assessment framework proposed by Hugonnet et al. (2022) to estimate uncertainties in our Topo DEMs.

Systematic biases in the difference of DEMs were addressed through coregistration and penetration correction using SRTM C-band data. Outlier removal further eliminated any remaining biases in glacierized areas.



Uncertainties in the corrected elevation differences stem primarily from two aspects including heteroscedasticity of elevation measurements and spatial correlation of errors (Hugonnet et al., 2022). Throughout the assessment, the median was employed as a robust estimator of central tendency, while the normalized median absolute deviation (NMAD) served as a measure of statistical dispersion.

$$NMAD = 1.4826 \cdot median(|dh_i - median(dh)|) \quad (4)$$

Where $dh$ denotes the full sample of per-pixel difference values between the Topo and SRTM DEM surfaces after co-registration.

Elevation heteroscedasticity was estimated by sampling the empirical dispersion of elevation differences. The NMAD was employed to quantify dispersion within binned categories defined by terrain slope and maximum absolute curvature. Based on this analysis, a functional relationship between the variance of elevation differences

($\sigma_{dh}$) and terrain slope ($\alpha$) and maximum absolute curvature ($c$) was established as $f = \sigma_{dh}(\alpha, c)$. The error for each pixel was estimated by using this function.

To mitigate the influence of heteroscedasticity on spatial autocorrelation analysis of residual errors in the Topo DEM products, we standardized the elevation differences using the error map. This procedure aligns with Equation 12 of Hugonnet et al. (2022) and enhances the robustness of error estimation by removing variance variability

caused by heteroscedasticity. Subsequently, we sampled empirical variograms (Equation 5) for standard elevation difference maps between the Topo and reference DEMs over stable surfaces using Dowd's estimator.

$$2\gamma_{dh}(d) = 2.198 \cdot median\left(z_{dh}(x,y) - z_{dh}(x',y')\right)^2 \quad (5)$$

Where $d$ is spatial lag between locations $(x,y)$ and $(x',y')$, $z_{dh}$ is the standard elevation difference, which can be calculated by $dh/\sigma_{dh}(\alpha,c)$.

For each difference map group, we employed a variogram analysis to characterize the spatial autocorrelation of residuals. This involved sampling and averaging 10 unique variograms, each with a sample size of 5000 pixels. we fit two nested variogram models to the empirical variograms using weighted least squares: a Gaussian model at short ranges and a spherical model at long ranges. This combination yielded the smallest least-squares residuals for most map groups. In some cases, a double-range nested spherical variogram model provided a better fit. Finally, the

spatially-integrated uncertainties ($\sigma_{\overline{dh}}$) for elevation difference within a specific area, such as a glacier, can be calculated using the following equation:

$$\sigma_{\overline{dh}} = \frac{\sigma_{dh}}{\sqrt{N}} \quad (6)$$

Where $\sigma_{dh}$ is vertical precision of the samples in the specific area, and $N$ is the number of independent pixels in the specific area based on the spatial correlations modelled by $\gamma_{dh}$. The solution for $N$ can be found by referring to the

methods section in the Supplementary material for Hugonnet et al. (2022).

**2.5.2 Uncertainties in mass balance estimation**

The annual surface elevation changes ($\frac{\partial dh}{\partial t}$) were converted to mass balance ($mb$) by using the Equation (7). A standard bulk ice density of $\rho$=850 kg m$^{-3}$ (Huss, 2013) was applied to assess the water equivalent.

$$mb = \frac{\partial dh}{\partial t} * \rho \quad (7)$$

The uncertainties included in the glacier area and glacier density should be taken into account in the estimation of the mass balance uncertainty. The uncertainty of glacier area is reported approximately 8% for the South Asia (Pfeffer et al., 2014), accounting for temporal evolution of glacier extent and glacier boundary mapping. Referring to study of Huss (2013), the density uncertainty can be assumed to $\Delta\rho$=60 kg m$^{-3}$. Considering all uncertainties to be uncorrelated, we can estimate the total uncertainty in mass balance using Equation (8).





$$\sigma M = \sqrt{\left(\frac{dh}{t} * \frac{\Delta\rho}{\rho_w}\right)^2 + \left(\frac{\sqrt{\sigma_{\overline{dh}}^2 + \left(\frac{\sigma A}{A}\right)^2}}{t} * \frac{\rho}{\rho_w}\right)^2} \quad (8)$$

Where, $dh$ is glacier thickness change, $t$ denotes the observation period, $\rho_w$ is the density of water (999.972 kg m⁻³), $\sigma A$ and $\sigma\rho$ are glacier area uncertainty and density uncertainty, respectively.

**2.6 Development of glacier-level and gridded mass change dataset**

Utilizing the final elevation difference data, we extracted the elevation change results for each glacier by clipping to the CGI1 glacier boundary. Subsequently, we filtered the glaciers to eliminate erroneous estimates of glacier elevation change due to incomplete topographic map coverage. To achieve this, we defined coverage ratios (R_acc & R_abla) for the accumulation and ablation zones, respectively. These ratios represent the proportion of coverage area of elevation difference compared to the total area of the accumulation or ablation zones of a glacier. The

accumulation and ablation zone for a specific glacier are identified by Equilibrium Line Altitude (ELA) which was defined as median surface elevation of a glacier referring to Sakai et al. (2015). Glaciers with coverage ratios of R_acc ≥ 0.4 (accumulation zone) and R_abla ≥ 0.5 (ablation zone) were classified as "effective glaciers" and included in the final glacier-level product and subsequent analyses. These thresholds were established to ensure sufficient data coverage within each zone for reliable estimates of glacier mass balance. In total, 13117 out of 24869

(~52%) glaciers met this criterion (~27% of glaciers lack any topographic map coverage). Notably, even though the number of effective glaciers represents a small percentage, their total area coverage accounts for ~72%. With the selection of effective glaciers, we proceeded to generate the initial glacier-level product. To facilitate the calculation of glacier mass balance, we created a time layer based on the acquisition date of each topographic map. This time layer accounts for potential differences between pixels within a glacier since a glacier may be divided into several

parts by Topo maps, albeit rare. In our products, the time layer distinguishes these differences. Additionally, within the attributes of each glacier elevation product, we provided the average elevation change for the glacier, the number of effective pixels ($N$) characterizing spatial autocorrelation and the final elevation change error $\sigma_{\overline{dh}}$ (computed based on section 2.5).

Compared with the glacier-level products, the primary challenge for gridded dataset lies in ensuring the

representativeness of the elevation difference within each grid rather than relying solely on coverage ratios. This means that a large number of samples are required for a specific grid, especially for grid with large size (e.g., 0.5°). However, most of the studies fall short of this requirement due to sparse image coverage across large regions. These studies often estimate glacier mass balance for a large-scale region using data from one or several small, glacierized area located within the broader region (e.g., Pieczonka and Bolch, 2015; Bhattacharya et al., 2021). Fortunately, the

comprehensive topographic map coverage employed in this study ensures a sufficient number of samples for producing the gridded product. To achieve gridded product generation, we follow the following steps: 1) We begin by spatially merging the elevation difference, elevation difference error, and time stamps for all effective glaciers. 2) For each grid cell, we calculate the average of the glacier elevation difference, elevation difference error, and time stamps within that grid cell. These averaged values are assigned as the grid cell values. 3) For each grid cell,

we compute the NMAD of elevation difference and the number of pixels. Using Equation (6), we calculate the grid-scale resampling error. 4) Combining the grid-scale elevation difference and time stamps, we apply Equation (7) to calculate the grid-scale mass balance. 5) Combining the grid-scale elevation change error and time stamps, we apply Equation (8) to calculate the initial grid-scale mass balance error. Finally, we incorporate the resampling error to obtain the final mass balance error.

The processing flow chart for the glacier-level and gridded datasets can be found in Fig. 2.

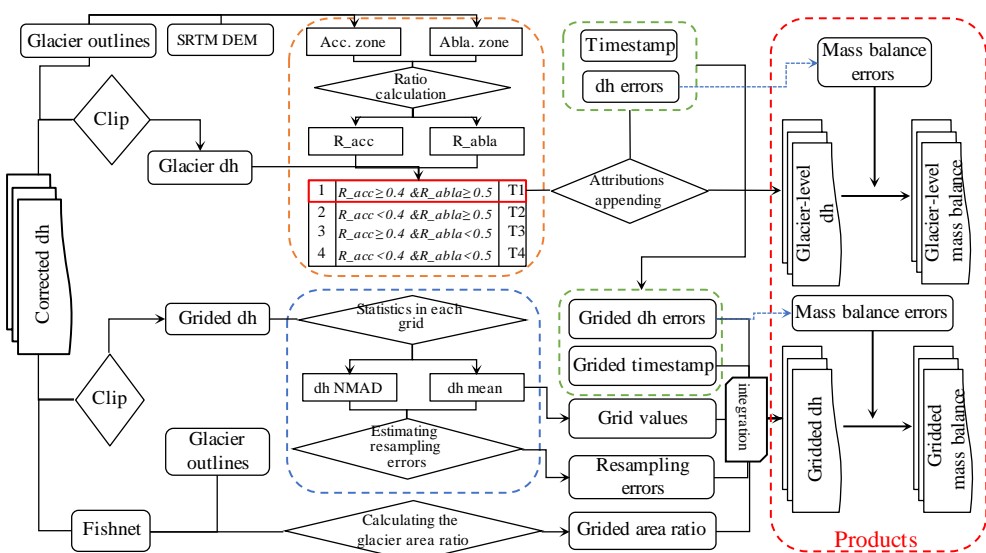

**Figure 2.** Workflow diagram for glacier-level and gridded product generation.

## 3 Results

### 3.1 Statistical uncertainties

At the glacier scale, the average error in glacier elevation difference is 4.16 m. Considering the time difference between the acquisition dates of the topographic maps, the average annual error in glacier elevation change for the entire study area is 0.14 m. Considering the uncertainty in glacier area and density, an average uncertainty of 0.12 m for the mass balance in the study area can be calculated using Equation (8). The average error in elevation difference for the 0.1° product is 3.91 m. This translates to an error of 0.13 m in glacier mass balance. For the 0.5° product, the average error in elevation change is 3.43 m, again corresponding to a 0.13 m error in mass balance. While there is a slight difference in error between the two resolutions, the overall results are relatively close. This difference may be attributed to statistical errors arising from resampling.

Statistical uncertainties exhibit significant spatial heterogeneity. The Ganges basin and the junction of the Brahmaputra and Salween basins show larger uncertainties, as shown in Fig. 3. This spatial variation is further supported by basin-level analysis of elevation difference uncertainty data (Table 2). The Irrawaddy basin exhibits the largest error, followed by the Ganges. These spatial differences can be partially explained by analyzing elevation difference for off-glacier areas. As illustrated in Fig. S4, the median and NMAD of elevation change reveal these variations. However, the primary contributor to this spatial heterogeneity is likely spatial autocorrelation in elevation change, particularly long-range correlation (Fig. S5).

Despite implementing the processing at the group level, glaciers within the same group exhibit varying errors. This variation arises from two key factors: inherent pixel-level errors associated with the elevation heteroscedasticity for each glacier, and the number of effective pixels calculated by spatial distances of long-range and short-range correlation used to represent each glacier (see dataset attributes).

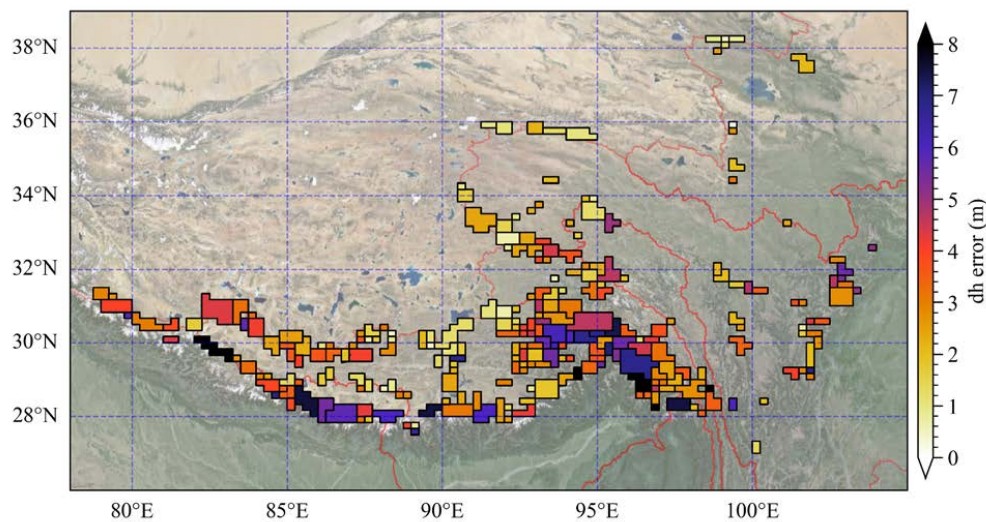


**Figure 3.** Spatial distribution of glacier surface elevation change error from 1970s to 200. The polygons represent group.

**Table 2** Statistics of vertical errors in each basin.

| Region | $median$(m) | $NMAD$(m) | $N$ | No. of glaciers | No. of Group | $\sigma_{dh}$(m) | $\sigma_{\overline{dh}}$(m) |
|---|---|---|---|---|---|---|---|
| Ganges | 1.02 | 12.99 | 14193 | 2055 | 17 | 10.70 | 5.37 |
| Brahmaputra | 0.59 | 14.61 | 92269 | 7322 | 147 | 11.92 | 4.36 |
| Salween | 0.51 | 12.69 | 25364 | 1792 | 26 | 11.22 | 3.74 |
| Mekong | 0.53 | 11.97 | 9055 | 358 | 5 | 11.82 | 3.46 |
| Yangtze | 0.40 | 8.70 | 27789 | 1288 | 49 | 9.33 | 2.81 |
| Yellow | 0.38 | 7.71 | 2726 | 238 | 13 | 7.98 | 2.76 |
| Irrawaddy | -0.20 | 41.65 | 280 | 64 | 4 | 15.60 | 8.07 |


### 3.2 Glacier-level mass change

The area involved in statistics accounts for 72% of total glacierised area. The average elevation difference for the period from 1970s to 2000 was -9.52 ± 4.16 m, corresponding to a mass balance of -0.30 ± 0.12 m w.e. yr[-1]. Small glaciers showed more negative mass change than large glaciers. As glacier size increases, the rate of surface thinning

diminishes. However, the thinning of larger glaciers is subject to greater uncertainty (Table 3). The uncertainties in surface thickness change range from 0.35~31.98 m for the study period. Detailed information on these uncertainties can be found in the glacier-level dataset. Large glaciers are becoming increasingly crucial as they are expected to persist for a longer duration compared to smaller glaciers, which are continuously disappearing due to the relentless effects of climate change. To gain a deeper understanding of elevation changes in large glaciers, we conducted

statistical analyses on glaciers with an area exceeding 10 km² (Fig. 4). The results revealed that the largest thickness reduction (~-13.4 m) occurred in the northeastward glaciers, which also hold the largest area proportion. Overall, a correlation was observed between the average slope of the glaciers and the magnitude of surface thinning. However, glaciers facing east and southeast exhibited a somewhat different pattern.

The Irrawaddy basin exhibited largest mass loss with values about -0.68 m w.e. yr[-1]. However, this estimate



also carries the large uncertainty (Table A3). The Ganges (-0.38 ± 0.19 m w.e. yr$^{-1}$) and Brahmaputra (-0.30 ± 0.13 m w.e. yr$^{-1}$) basins, which collectively account for over 80% of the glacier coverage in the ETPR, exhibit mass balance trends that closely reflect the pattern for the entire study area. The Yellow River basin, on the other hand, displayed the smallest relative error in its mass balance estimate (-0.19 ± 0.06 m w.e. yr$^{-1}$), suggesting higher quality terrain data coverage in this region.

**Table 3.** The statistics of elevation difference (dh) for different glacial scale. The standard scale range for glaciers refers to Shi et al. (1988).

| Area range (km$^2$) | No. of glaciers | Glacier area (km$^2$) | No. involved in statistics | Area involved in statistics (km$^2$) | dh (m) | dh error (m) |
|---|---|---|---|---|---|---|
| ≤0.5 | 15108 | 3147.03 | 6533 | 1549.86 | -10.76 | 4.97 |
| 0.5-1 | 4145 | 2924.00 | 2566 | 1828.80 | -9.87 | 3.52 |
| 1-5 | 4561 | 9457.06 | 3216 | 6771.28 | -9.28 | 3.25 |
| 5-10 | 602 | 4162.66 | 451 | 3114.89 | -8.08 | 3.47 |
| 10-30 | 358 | 5595.20 | 273 | 4258.72 | -7.30 | 3.30 |
| 30-50 | 62 | 2297.10 | 50 | 1842.20 | -5.65 | 3.39 |
| 50-80 | 23 | 1423.61 | 19 | 1183.21 | -2.90 | 3.27 |
| 80-100 | 5 | 458.52 | 5 | 458.52 | -5.11 | 4.23 |
| 100-300 | 5 | 809.06 | 4 | 687.17 | -4.44 | 4.56 |

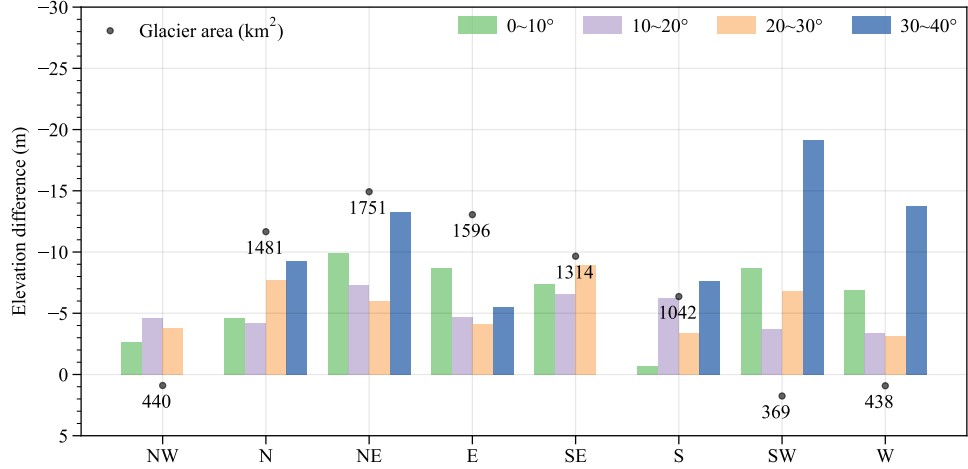

**Figure 4.** Elevation difference of large glaciers (> 10 km$^2$) by aspect and slope.

### 3.3 Gridded glacier mass balance

This study presents high-resolution (0.1° and 0.5°) gridded products of glacier elevation change for the ETPR. These products offer significant advantages for analyzing spatial patterns of glacier mass balance. The gridded format allows for efficient identification and analysis of "hotspot" regions experiencing significant glacier elevation changes. Moreover, they serve as a critical foundation for characterizing glacier runoff in glacierized basins and calibrating parameters in hydrological models that simulate meltwater contribution to total runoff. Glacier in the ETPR exhibited a mass balance of -0.29 m w.e. yr$^{-1}$ (corresponding to -9.23 m surface thinning during study period) at a 0.5° grid scale for the investigated period (Fig. 5). Our analysis reveals minimal and statistically insignificant



differences in mass balance estimates due to grid size variations. For example, the 30 m resolution product yielded a surface thinning value of -9.52 m, compared to -9.41 m obtained from the 0.1° grid product. These negligible discrepancies are likely attributable to the resampling process used to generate the gridded data. Details on resampling procedures are provided with each product for further reference. Our analysis revels an apparent discrepancy in glacier mass balance by the meridional and zonal distribution. Specifically, there is a decrease in mass balance with increasing latitude, and two hotspot regions (the source area of the Brahmaputra and the upper reaches of the Mekong, Salween, and Irrawaddy) exhibit the largest negative mass balance along the meridional direction.

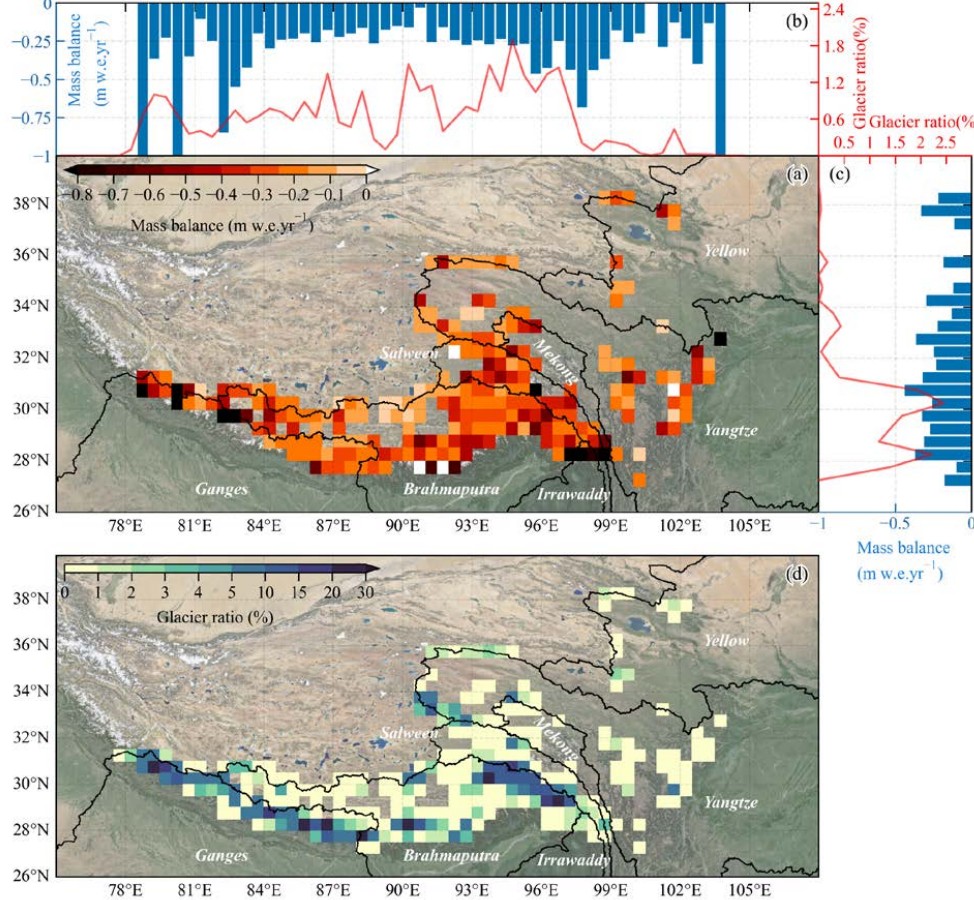

**Figure 5** 0.5° gridded product over the ETPR. (a) Characteristics of glacier mass balance. (d) Glacier ratio. Statistics of mass balance along longitude and latitude are presented in (b) and (c), respectively. The glacier ratio accounts for the glacier area divided by grid area.

## 4 Discussions

### 4.1 Evaluation of Topo DEM based elevation difference

Given the confidentiality and unavailability of topographic maps, directly quantifying the systematic and random errors introduced during map scanning, distortion correction, and DEM generation processes proves exceptionally





challenging. Consequently, utilizing stable terrain as a proxy to evaluate the reliability of the Topo DEM has emerged as an alternative approach. This approach involves a comparative evaluation with a DEM generated from

KH-9 data acquired around the same temporal window. The well-established credibility of KH-9 data, coupled with its documented application in various research in HMA (e.g., Zhou et al., 2018; King et al., 2019), strengthens the validity of this comparative strategy.

The Xixiabangma Mountain range was chosen as the test site due to its near-contemporaneous acquisition times (November 1974) for both datasets and its complex topography, representative of the wider ETPR. To

minimize the influence of errors introduced during elevation difference extraction, identical processing workflows were applied to both DEMs (details provided in Section 2.4), with the exception of the DEM generation method itself. Statistical analyses were performed on approximately 2.8 million off-glacial pixels (Fig. 6). The analysis revealed a broadly similar pattern in elevation differences below 6000 m a.s.l. for both DEMs. However, discrepancies emerged at higher elevations (above 6000 m a.s.l.). In the 6000-7000 m a.s.l. range, the Topo DEM

exhibited an elevation difference close to 0, while the KH-9 DEM showed a positive difference. Notably, the large standard deviation (>40 m) in both cases indicated a low degree of reliability for elevation differences at high altitudes. To quantitatively assess the similarity, we employed histogram statistics. The median and NMAD of the off-glacial elevation differences were calculated before and after the coregistration. The results (Fig. 6e & f) demonstrate a more concentrated distribution centred around zero and a smaller NMAD for the Topo DEM based

elevation difference compared to the KH-9 DEM based elevation difference at stable areas. It suggests that Topo DEM implies a higher level of product quality compared to the KH-9 DEM in this region, but this warrants further validation through additional comparative studies and a more comprehensive error characterization of both DEMs.

We therefore evaluated the precision of the KH-9 and Topo DEMs by examining two aspects: heteroscedasticity and spatial autocorrelation. The analysis revealed a pronounced dependence of errors on both

slope and curvature for both the topographic map and KH-9 data. In areas with a maximum absolute curvature less than $1/100\ \text{m}^{-1}$ and a slope less than 20°, the NMAD of the off-glacier region for the Topo DEM was 4.7 m, while the KH-9 DEM exhibited a significantly higher NMAD of 13.4 m (Fig. 7). Although both DEMs displayed larger errors in high-altitude, steep slope regions, the errors associated with the KH-9 DEM were substantially greater, which aligns with the previous statistical findings. This suggests that the Topo DEM exhibits lower errors stemming

from elevation heteroscedasticity. The spatial autocorrelation evident in Fig. 8 was further quantified by analyzing both short-range and long-range correlation. The results indicated that the Topo DEM consistently displayed lower values for both types of autocorrelation compared to the KH-9 DEM. Notably, the long-range autocorrelation, which exerts a greater influence on the uncertainty of glacier surface elevation changes (refer to Equation (5) and the supplementary of Hugonnet et al. (2022)), exceeded 30 km for both DEMs. This limited the number of independent

elevation difference pixels, ultimately increasing the uncertainty in the final estimates of glacier surface elevation change according to Equation (6). The observed issues with the Topo DEM likely stem from the scarcity of national elevation control points in border regions, leading to a comparatively lower DEM accuracy.

In addition, the concordance observed between Topo DEM and ICESat-derived elevation differences further corroborates the applicability of the Topo DEM product at low elevations. These findings suggest that the precision

of glacier surface elevation change results generated from topographic maps is, at minimum, on par with the established accuracy of glacier change products derived from KH-9 data. However, it is imperative to acknowledge the limitations associated with Topo map products. When employing these data for specific applications, it is essential to consult the uncertainty assessments provided within our dataset for each individual glacier or grid cell.
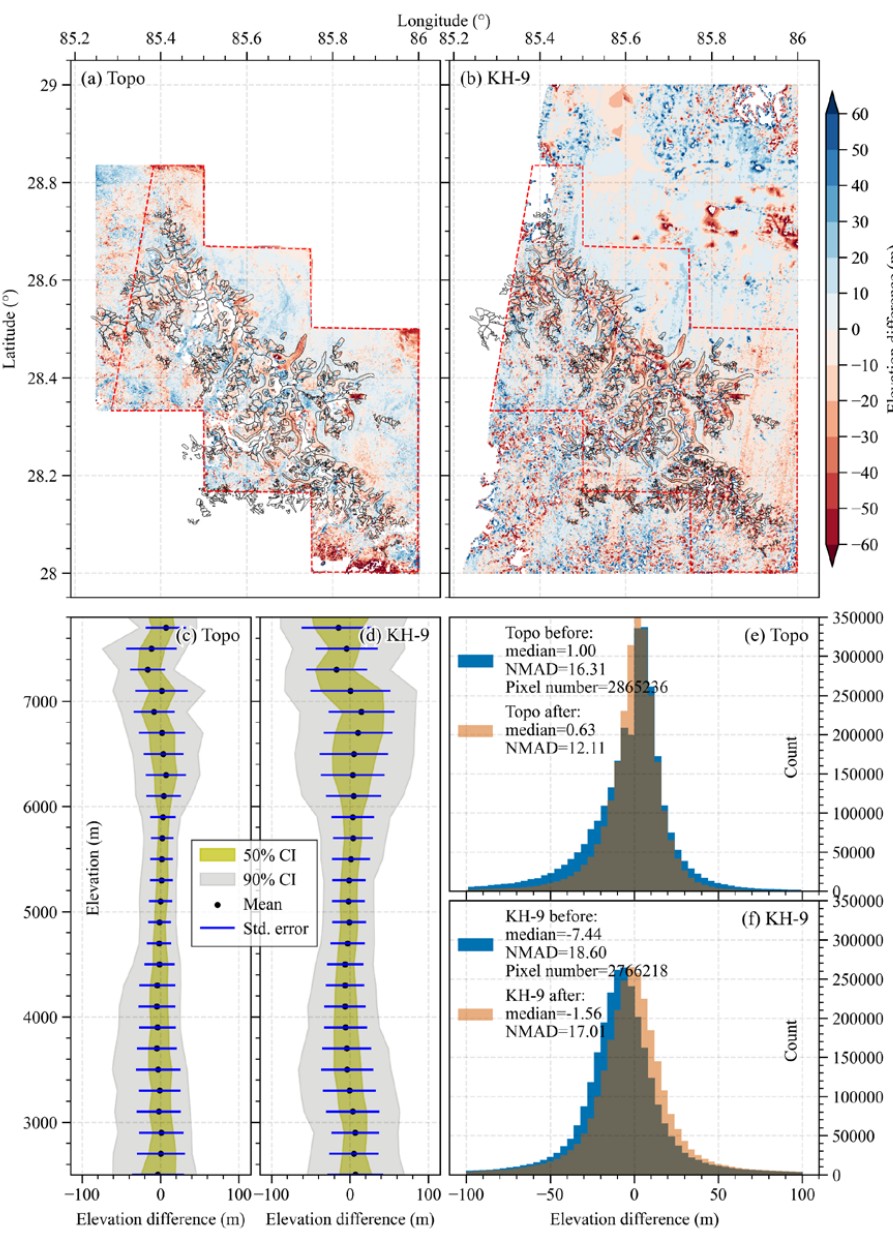


**Figure 6.** Comparison of (a) Topo based and (b) KH-9 based elevation difference for the off-glacier region. The statistics on elevation difference along elevation for Topo and KH-9 results were displayed in (c) and (d). Frequency Distribution of elevation differences for Topo DEM and KH-9 DEM is presented in (e) and (f), respectively. The CI denotes the confidence interval.






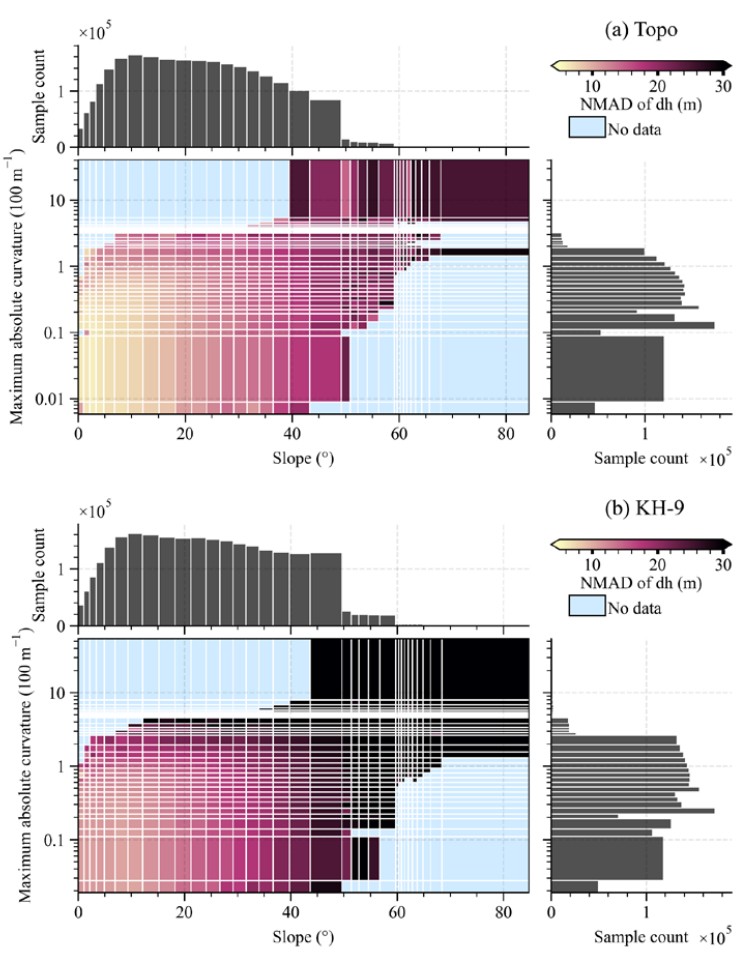

**Figure 7.** Statistical analysis of off-glacier elevation differences in Topo DEM (a) and KH-9 DEM (b) as a function of maximum absolute curvature and slope.

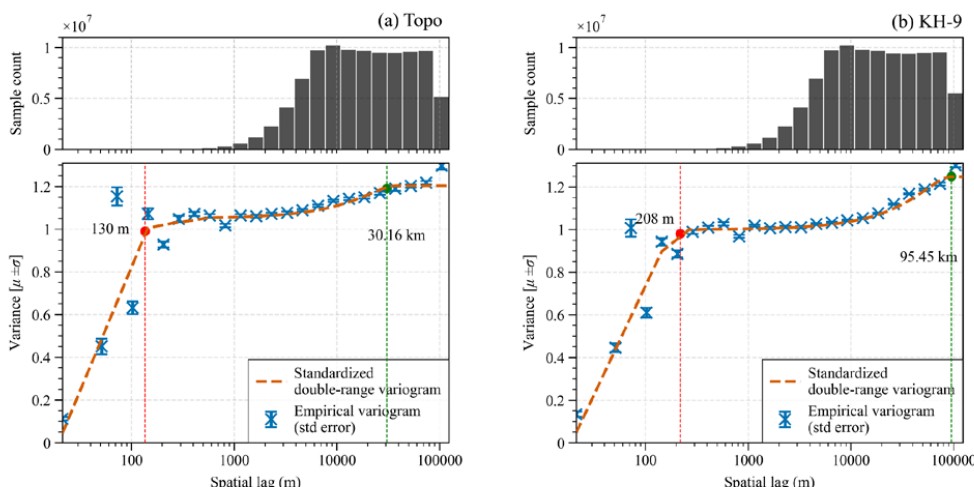

**Figure 8.** Spatial autocorrelation characteristics of Topo DEM (a) and KH-9 DEM (b). For both data, a gaussian model was used to fit short-range variogram, while a spherical model for long-range variogram.

### 4.2 Uncertainties arising from topographic map production processes

Most of uncertainties in Topo DEMs stem from the process of the generation of the contours and the interpolation (Junfeng et al., 2018). Achieving high map accuracy requires adherence to standard contour intervals (see Table A1). However, it is increasingly difficult in steep terrains because dense lines cannot be draw within the limited space (Fig. A2). This coupled with the strategies in cartographic generalization of diverse land cover, lead to uncertainties in the contours.

To assess the uncertainties, we analyzed off-glacier regions categorized into six land cover types (Fig. 9). Grassland and bare land exhibited minimal uncertainties (~0.31 m and -0.44 m, respectively) due to their minimal elevation variations compared to forested areas. Additionally, their larger and more continuous high-altitude patches facilitated denser elevation point placement. Notably, a positive difference (~1.21 m) arose from neglecting tree height in contour generation. Snow cover regions displayed increasing uncertainty with elevation and slope. Despite comprising approximately 8% of the whole statistical region, its uncertainty mirrored that of glacial area due to similar distributions. High contour line density is crucial for accurate mapping in steep high-altitude terrains. However, mapmaker limitations and national standards specifying wider spacing in high-altitude regions (Fig. A2 & Table A1) lead to significant uncertainties in snow-covered areas. Low-altitude snow cover variations minimally impact long-term glacier elevation changes. Due to dispersed distributions of water bodies and the expansion of the lakes during study period, the elevation difference in water bodies is inaccurate. Uncertainty in orientation remained consistent across land cover types (Fig. 9c). On the whole, uncertainties increase with slope, likely due to past compilation limitations and poorer technical capabilities in representing steep terrains. This slope-dependent accuracy issue is common to other DEM products (SRTM DEM, ASTAR GDEM, SPOT5-based DEM) (Kocak et al., 2004; Rodríguez et al., 2006). Consequently, geodetic-based ice mass change estimates might have significant uncertainties in steep terrains, particularly glacier accumulation zones.

Another uncertainty is the interpolation in generating DEM from the contours. Interpolation methods have a certain amount of influence on accuracy and natural neighbour method displays a relative high accuracy (Habib,



2021). However, almost every interpolation method faces challenges in high altitude regions since the interpolation leads to a smooth surface structure, especially in steep slopes. Despite the meshing techniques can be used to establish a refined topographical structure which is more closed to real terrain (Pieczonka et al., 2011), the limited
number of elevation points (8-15 for per 100 mm$^2$ in alpine region in topographical maps according to GB/T 12343.1-2008) makes it hard to conduct this procedure when generating Topo DEMs. In spite of the difficulties, there are numerous successful application research on glacier changes based on topographical maps (e.g., Ye et al., 2017; Junfeng et al., 2018; Wu et al., 2018).

        Nevertheless, we undertook a comprehensive quantitative assessment of these uncertainties by analyzing the
elevation difference characteristics of off-glacial regions (refer to Section 3.1). In contrast to previous studies that relied on topographic maps (e.g., Ye et al., 2017; Wu et al., 2018), our approach quantifies the long-range spatial autocorrelation errors, potentially arising from the topographic map generation process, rather than solely focusing on short-range spatial autocorrelation errors introduced by interpolation. This more comprehensive error assessment strengthens the robustness and applicability of our data.

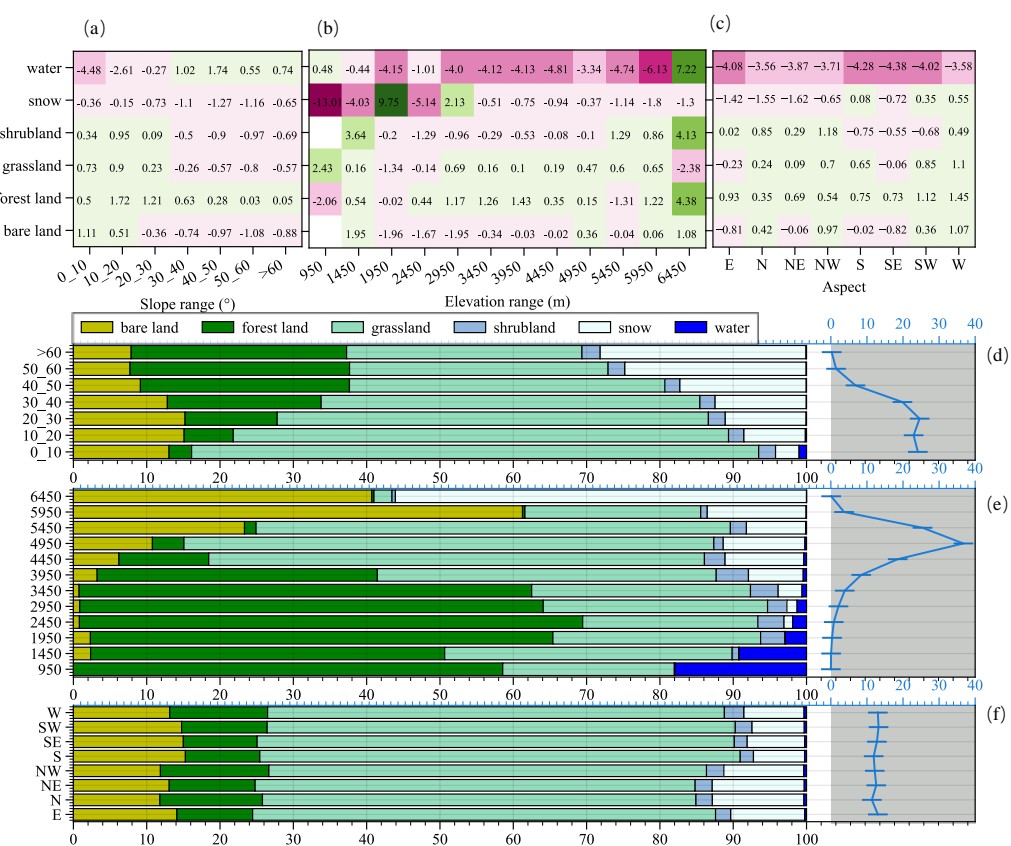


**Figure 9.** Elevation differences in off-glacier regions by slope (a), elevation zone (b), and aspect (c) across different land covers. The land cover ratios are displayed in (d), (e), and (f). Blue lines within gray shadows in (d), (e), and (f) represent the total area proportions in a specific classification. The units in (a), (b), and (c) are m, while in (d), (e), and (f) are %.


**4.3 Comparison on glacier mass changes derived from Topo and KH-9**

To date, there is little observation for mass changes prior to 2000 in the whole ETPR. Few regional studies based on topographical maps (e.g., Ye et al., 2015) offer some insights, but their coverage is limited. Conversely, studies utilizing KH-9 data (e.g., Maurer et al., 2019; Bhattacharya et al., 2021) with sufficient coverages can serve as an independent source for comparison and validation of mass balance derived from Topo DEM. We therefore compared our dataset with the KH-9 assessments, particularly the works of Zhou et al. (2022) and Bhattacharya et al. (2021). Focused on the Himalayas, previous studies reported mass changes ranging from -0.17 to -0.32 m w.e. yr$^{-1}$ (Table A2) generally consistent with our findings. However, some discrepancies emerged at the basin or regional scale. For example, the Yigong Zangbo and Parlung Zangbo (mainly included in the Salween and the Brahmaputra) were reported a mass balance of -0.11 ± 0.14 m w.e. yr$^{-1}$ and -0.19 ± 0.14 m w.e. yr$^{-1}$ respectively by Zhou et al. (2018). These values diverge slightly from our Topo DEM-derived estimate of less than -0.24 ± 0.12 m w.e. yr$^{-1}$, potentially due to incomplete data used in the previous study. This difference highlights the significance of data source and methodology for the estimation of mass balance. As other datasets were unavailable, we used the results of Bhattacharya et al. (2021) for further analysis.

Our total ETPR-wide mass balance (-0.30 m w.e. yr$^{-1}$) is slightly higher than value (-0.22 m w.e. yr$^{-1}$) reported by Bhattacharya et al. (2021). This difference could stem from sources. First, discrepancies in the methodologies used to generate DEMs from optical imagery might contribute. Second, a minor inconsistency exists in the reference periods for calculating elevation changes. The acquisition dates for KH-9 images primarily range from 1964-1976, while topographical maps were acquired from 1957-1983. While we accounted for annual differences, slight variations due to these timeframes might persist. However, the most important factor likely influencing the difference is the coverage of the compared results. As Wang et al. (2019) highlight, glacier mass balance exhibits significant spatial heterogeneity across large regions due to variations in topography and meteorological conditions. This heterogeneity can even be observed within specific mountain ranges. Consequently, it's challenging to determine whether regional mass balance estimates, based on a limited area, accurately represent the mass budget of the entire ETPR. Therefore, rather than focusing solely on the overall difference, a more meaningful analysis would involve comparing mass balance estimates within specific regions where both KH-9 and Topo DEM data overlap.

Due to the availability of the KH-9 elevation difference, only two sub-regions in the Ganges basin and one sub-region in the Brahmaputra basin were selected for regional comparisons. Notably, both two results revealed the similar heterogeneity in mass changes, with values increasing from east (Western Nyainqentanghla) to west (Gurla Mandhata). In the Western Nyainqentanghla, our estimate (-0.13 ± 0.06 m w.e. yr$^{-1}$) was less negative than the estimate based on the KH-9 (-0.24 ± 0.12 m w.e. yr$^{-1}$) during 1976-2001 (Bhattacharya et al., 2021). This could be attributed to the slightly different acquisition times. By comparing the elevation differences in the sub-region (Langtang) of the Poiqu (Fig. 10d and e, Table 4), we conclude that the difference of the data coverage and glacier boundary is also a trigger for less negative mass change in our results. The more negative mass balance derived from Topo DEM for the Gurla Mandhata region likely arises from data-filling strategies. The filled region (6250-6900 m) in our data exhibits a near-zero mass budget, while the corresponding area in the KH-9 data shows a gap (Fig. 10a & b). In order to clarify this phenomenon and promote more flexible applications of our dataset, we also provide the elevation difference data with no gap filling.

For all compared regions, the annual elevation differences display similar pattern across the altitude between KH-9 and Topo, and a notable discrepancy emerges at the glacier tongue (Fig. 10c, f, and i). Here, KH-9 data consistently yields more negative elevation changes (greater thinning rates) compared to Topo DEM data, with differences ranging from 0.2 to 0.5 m/yr. This discrepancy is particularly pronounced in the Nyainqentanglha region. It's important to note that our analysis excludes a small portion of the glacier tongue where the largest surface



thinking is observed. Despite this exclusion, our Topo DEM-based results exhibit relatively low variability (normalized median absolute deviation, NMAD, of 5.61 m) across elevations. This stands in contrast to the more pronounced variation seen in the elevation change distribution derived from the KH-9 data. These observations suggest potential uncertainties associated with either the production of Topo DEMs or KH-9 DEMs.

To further confirm that our data can give a comprehensive and effective expression in the whole study region,
we conducted a comparison with the results post 2000 (e.g., Brun et al., 2017; Shean et al., 2020; Hugonnet et al., 2021) which have a more extensive coverage. All studies, including our Topo DEM-based analysis, identified the Salween source area as a central zone of significant mass loss (Table A3). However, the results derived from KH-9 indicated that the east-central Himalaya was the region with largest glacier surface thinning. Under the assumption of relatively consistent climatic variability across basins, the spatial heterogeneity of mass balance at the basin scale
would likely remain same. By comparison, the Topo-derived mass balance was found to exhibit the same spatial heterogeneity with the average of the estimates of Brun et al. (2017) and Shean et al. (2020) (excluding the Yellow and Irrawaddy basins due to small glacier cover). This highlights how dataset coverage can influence our understanding of glacier mass balance heterogeneity for a large region.

**Table 3** Glacier mass balance derived from KH-9 and Topo DEMs in selected regions. Source of the results derived from KH-9 panoramic camera data(Bhattacharya et al., 2021)

| Region | KH-9 | Topo | Periods (KH-9 vs. Topo) | Referred watershed |
|---|---|---|---|---|
| Western Nyainqentanghla | -0.24±0.13 | -0.19±0.07 | 1976-2001 vs. 1974-2000 | Brahmaputra |
| Poiqu | -0.30±0.10 | -0.21±0.13 | 1974-2004 vs. 1970-2000 | Ganges (east) |
| Gurla Mandhata | -0.12±0.10 | -0.13±0.08 | 1974-2004 vs. 1974-2000 | Ganges (west) |





**Figure 10.** Annual elevation difference derived from Topo and KH-9. The distribution of elevation difference along altitude is presented in (c), (f), and (i). The light shadows in (c), (f), and (i) represent the 50% CI.

## 5 Conclusions

This study presents a near-complete glacier mass change dataset for the pre-2000 period, encompassing approximately 70% of the study area's total glacier extent. This comprehensive inventory provides valuable insights into the spatial heterogeneity of glacier changes across the region. The gridded products, in combination with the published results, provide a nearly 5-decades record of mass balance to support parameter calibration in hydrological simulation and energy balance simulation, and to evaluate the contribution of mountain glacier loss to sea level.

There overall mass balance in the ETPR from 1970s to 2000 is -0.30 ± 0.12 m w.e. yr$^{-1}$. Small glacier experienced the most negative mass change. The mass balance exhibits a pronounced latitudinal gradient, showing that regions farther south experience greater mass loss.

Compared to prior Topo map products, our datasets offer a more comprehensive analysis of glacier mass balance uncertainties by incorporating data from stable areas, addressing uncertainties from heteroscedasticity of elevation measurements, and considering the previously neglected uncertainties of long-range correlation in elevation measurements. This in-depth analysis pinpoints the generation process of elevation contours and the increasing uncertainty with steeper slopes as the main drivers of mass budget uncertainty. These findings highlight the importance of meticulously evaluating and mitigating these uncertainty sources when interpreting glacier mass balance estimates, especially in areas with challenging topography.

Systematic comparisons between Topo and KH-9 derived results indicate that our product achieves an accuracy level comparable to, if not exceeding, that of KH-9. However, both Topo and KH-9 based results give a low reliability at high altitude. Despite this, the overall mass balance estimates derived from Topo DEM data remain consistent with those obtained using KH-9. Nevertheless, some regional discrepancies in mass balance knowledge are evident when compared to KH-9 findings. These discrepancies can be attributed to two primary factors. Firstly, the inherent limitations in the spatial coverage of KH-9 data potentially restrict the comprehensiveness of the analysis in certain regions. Secondly, the specific methodological approaches employed during the generation of each DEM may introduce systematic biases that contribute to the observed variations.

## 6 Data availability

The ETPR glacier surface mass change database is distributed under the Creative Commons Attribution 4.0 License. The data can be downloaded from the data repository of the National Tibetan Plateau/Third Pole Environment Data Center at https://doi.org/10.11888/Cryos.tpdc.301236 (Liu et al., 2024).

## Appendix A

### A1 Comparative analysis with Topo DEM with ICESat-2 in off-glacier region

ICESat-2 ATL06 data (October 2018 - February 2019) was used to further evaluate the Topo DEM in stable off-glacier regions (Fig. A1). ATL06 boasts high accuracy, with horizontal and vertical uncertainties of less than 10 m and 3 mm, respectively (Smith et al., 2020; Brunt et al., 2019). While the mean elevation differences between ATL06-SRTM and SRTM-Topo were close to zero below 5700 m a.s.l (as shown in Fig. A1b), the ATL06 product exhibited a more discrete distribution. This likely stems from matching errors that arise when comparing laser points with 30 m resolution SRTM DEM. The findings suggest that the corrected Topo DEM differences exhibit comparable accuracy to ATL06 at these lower elevations. However, at higher elevations (above 5700 m a.s.l), negative elevation differences were observed in the Topo DEM. These differences are likely related to variations in



snow cover, influenced by both seasonal snowfall and potentially atmospheric warming. The significant changes in
the off-glacier region at high elevations limit the ability to definitively evaluate the reliability of the Topo DEM
using ATL06 data.

It's important to note that despite good agreement with other elevation data sources, uncertainties remain in the
Topo DEM at high altitudes (above 5700 m a.s.l.). This limitation is a common challenge for DEMs derived from
historical optical imagery (e.g., Zhou et al., 2018). Factors contributing to this limitation include the low accuracy
of the photographic equipment used, a lack of ground control points for precise georeferencing, and seasonal
variations caused by random shooting times.

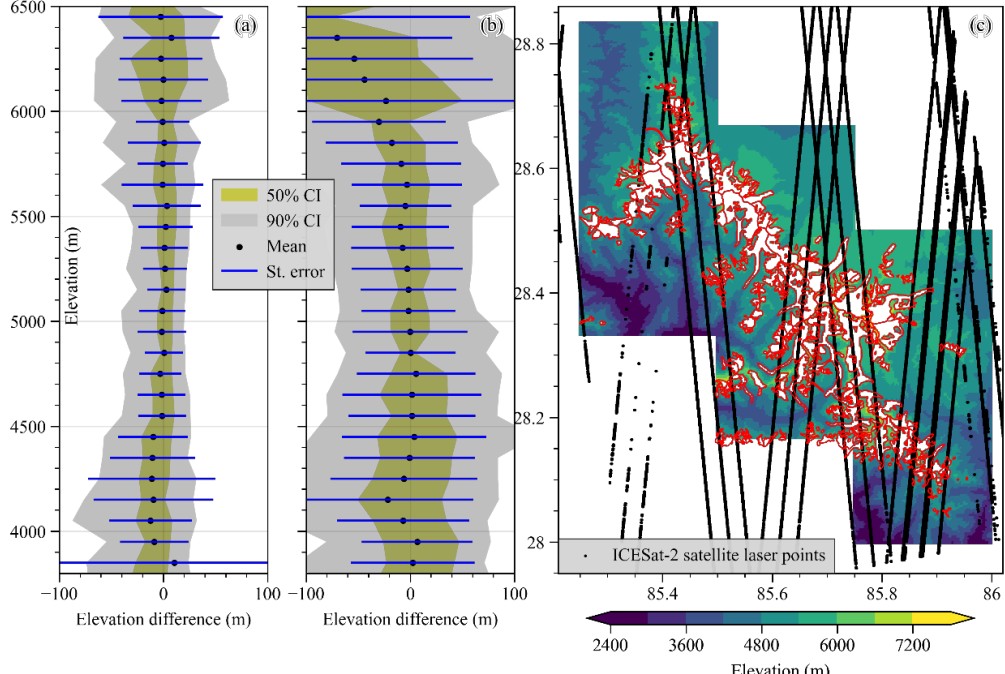

**Figure A1.** Comparison of Topo DEM (a) and ATL06 (b) elevation differences along altitude (Off-glacier region,
Xixiabangma Mountains). Tracks of ATL06 are presented in (c). Elevation zones with less than 200 laser spots of
ATL06 were excluded from the statistical analysis.

## A2 Standards in generating Topo maps

**Table A1.** The standard for generating contours from aerial photos (NPSC, GB/T 12343.1-2008).

| Landform | Basic contour distance (m) | |
|---|---|---|
| | 1 : 50000 | 1 : 100000 |
| Flat ground (Slope < 2° & elevation difference < 80 m) | 10 (5) | 20 (10) |
| Hill area (2° ≤ Slope < 6° & 80 m ≤ elevation difference < 300 m) | 10 | 20 |
| Mountainous region (6° ≤ Slope < 25° & 300 m ≤ elevation difference < 600 m) | 20 | 40 |
| Alpine region (Slope > 25° & elevation difference > 600 m) | 20 | 40 |



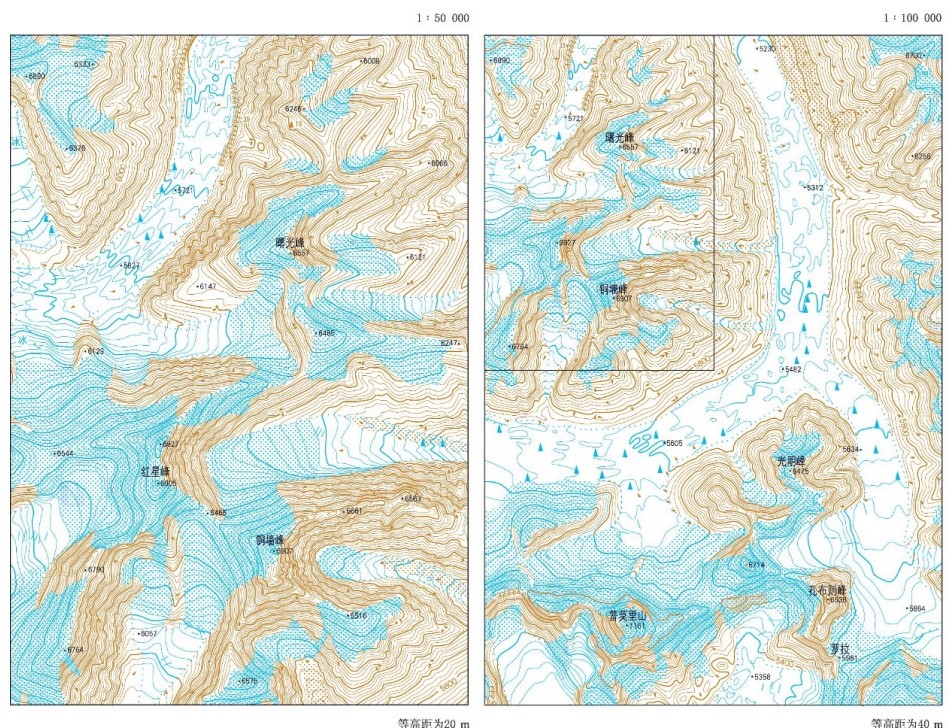

**Figure A2.** A sample of topographical maps in glacierized region (originate from the NPSC, GB/T 12343.1-2008).

## A3 Previous studies on mass balance in ETPR

**Table A2.** The previous studies using 1970s KH-9 Metric Camera imagery in the study area.

| Region | Mass balance (m w.e. yr$^{-1}$) | Period | Reference |
|---|---|---|---|
| Himalayas (25° - 37°N, 75° - 93°E) | -0.22 ± 0.13 | 1975 - 2000 | Maurer et al. (2019) |
| Himalayas (25° - 37°N, 75° - 93°E) | -0.25 ± 0.09 | 1974 - 2000 | King et al. (2019) |
| Langtang (28.10°-28.39°N, 85.50° - 85.80°E) | -0.24 ± 0.08 | 1974 - 2006 | Ragettli et al. (2016) |
| Gangotri Glacier, Garhwal (30.802°N, 79.147°E) | -0.20 ± 0.01 | 1968 - 2006 | Bhattacharya et al. (2016) |
| Eastern Himalayas (27.90°- 28.35°N, 89.80° - 90.90°E) | -0.17 ± 0.05 | 1974 - 2006 | Maurer et al. (2016) |
| Everest area (27.83°-28.17°N, 86.75° - 87°E) | -0.28 ± 0.12 -0.35 ± 0.12 | 1962 - 2001 2001 - 2018 | King et al. (2020) |
| Everest area (27.83°-28.17°N, 86.75° - 87°E) | -0.32 ± 0.08 | 1970 - 2007 | Bolch et al. (2011)* |

\* denotes studies using the Corona KH-4





**Table A3.** Statistics for the mass balance from Shean's and Brun's Study. The units of mass balance (MB) and uncertainties are m w.e. yr$^{-1}$

| Region | This study (1970s-2000) | | Shean's (2000-2018) | | Brun's (2000-2016) | | Average (Shean & Brun's) | |
|---|---|---|---|---|---|---|---|---|
| | MB | Error | MB | Error | MB | Error | MB | Error |
| Ganges | -0.38 | 0.19 | -0.34 | 0.27 | -0.31 | 0.44 | -0.33 | 0.36 |
| Brahmaputra | -0.30 | 0.13 | -0.42 | 0.37 | -0.43 | 0.46 | -0.42 | 0.41 |
| Salween | -0.26 | 0.10 | -0.52 | 0.30 | -0.63 | 0.49 | -0.57 | 0.40 |
| Yellow | -0.19 | 0.06 | -0.46 | 0.26 | — | — | -0.46 | 0.26 |
| Yangtze | -0.22 | 0.08 | -0.40 | 0.27 | -0.35 | 0.33 | -0.37 | 0.30 |
| Mekong | -0.29 | 0.09 | -0.45 | 0.31 | -0.49 | 0.41 | -0.47 | 0.36 |
| Irrawaddy | -0.68 | 0.33 | -0.59 | 0.44 | -0.48 | 0.57 | -0.53 | 0.50 |


**Author contributions**. SL designed the framework of the mass change database. YZ programmed the co-registration algorithms, produced the two mass change datasets, and wrote the manuscript. SL edited the first manuscript. TB reviewed and improved the manuscript. JW, KW, JX, WG, ZJ, FX, YY, DS, XY and ZZ adjusted and transformed the digitized contour maps and produced the Topo DEM. All authors discussed and improved the
manuscript.

**Competing interests.** The contact author has declared that none of the authors has any competing interests.

**Acknowledgments**. We thank the Chinese Military Geodetic Service for providing the digitized contour maps.

**Financial support**. This research has been supported by the National Natural Science Foundation of China (Grant No. 42171129, 42301154) the International Science and Technology Innovation Cooperation Program of the State Key Research and Development Program (Grant No. 2021YFE0116800, 2023YFE0102800), the Research Project of Postdoctoral Research Fund of Yunnan Province (Grant No. C615300504038).

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
