# Peer review of "Glacier-level and gridded mass change in the rivers' sources in the eastern Tibetan Plateau (ETPR) from 1970s to 2000"

_Earth System Science Data, 2024_

## Author Response (AR2)

We thank the reviewers for the constructive comments and suggestions. The comments are helpful and will certainly strength the quality of the manuscript. Our responses to the comments are highlighted in blue font. All specific modifications have been made in the revised manuscript for your next round of review. It should be noted that because the system does not support uploading the supplementary material with tracked revisions, there are no tracked changes in our revised supplementary material.

**Remarks from the preceding review file validation**

Since the base map in figures 3, 5, and S2 is from Google, please add "© Google YEAR" to the map or at least to its caption during the next revision.

Thank you for your kind reminder, we have added "© Google 2024" to the captions of the figures as suggested.

**Response to suggestions from Topic Editor**

Thank you for your revisions. I'll ask reviewer#1 to review your manuscript to check if your corrections are satisfying. I suggest you move the Appendix to the supplementary materials, as it seems redundant to have both.

Thank you for your suggestions. We moved the Appendix to the supplementary materials and made the corresponding adjustments in the main text.

**Response to Reviewer #1 (Dr. Romain Hugonnet)**

General

I reviewed a previous ESSD submission of this study (https://essd.copernicus.org/preprints/essd-2022-473/).

This manuscript by Zhu et al. presents a glacier elevation change and mass change estimate for the eastern Tibetan Plateau. This dataset is based on historical topographic maps and spy satellite imagery from the 70s, compared to the Shuttle Radar Topography Mission of February 2000. There are little glacier mass change estimates before the 2000, and hence the main added value of this study is to extend the observational period of glacier mass change back to the 70s.

Overall, this is an interesting study that has greatly improved since the past submission that I reviewed. I still have a couple of main comments, less crucial than the ones of my previous review yet still important, as well as a couple of other minor remarks described below.

Improvement since last submission

The authors have satisfactorily addressed most of the main concerns from my previous review: They introduce an additional penetration correction for the X-band instead of neglecting it (based on Zhou et al., 2018), They have revisited their uncertainty analysis to account for the long-range correlation of errors that are common in both KH-9 DEMs and SRTM DEMs (based on Hugonnet et al., 2022), and thus crucial to propagate errors to glacier mass changes, They have removed their elevation-dependent correction due to the biases it can create for glacier mass balance estimation (based on Gardelle et al., 2012), Additionally, the authors seem to have revised several statements and

references that were either convoluted or not relevant.

Thank you for your recognition of our revisions. The error evaluation tools you publicly shared have helped us tremendously. We will continue to refine our manuscript based on your feedback in this round.

Main comments

1/ Report confidence level

This was a remark in my past review. Maybe I have missed it, but I couldn't find the confidence level in the text. The authors should specify if their reported uncertainties throughout the paper refer to +/- 1 sigma (68% confidence level) or +/- 2 sigma (95% confidence level).

Thank you for your comments. Before we used xdem to calculate the error, we used +/- 2 sigma. Currently, our results use +/- 1 sigma. In the revised version, we have modified the error of mass balance to +/- 2 sigma and explained this change in the methods section.

2/ Grid-level aggregation

It is my understanding that regional hypsometric interpolation was performed in this study (line 197; missing reference to McNabb et al., 2019), yet it is directly followed by the statement "In our data products, we have preserved the elevation changes without applying interpolation". I'm not sure I understand: Does the above statement only refer to elevation change maps? Surely this interpolation is used for glacier mass change estimates, otherwise it would be useless. Please clarify. This clarification also joins the topic of grid-level aggregation: In this section, glaciers are discarded based on their ratio of valid data in ablation or accumulation zones. Does this only concern the non-interpolated elevation changes?

Additionally, the most crucial point for grid-aggregation is that it seems that the authors clipped elevation change maps by their desired tile size (0.1°x0.1° or 0.5°x0.5°), then reproduced the same computations done at the glacier level.

The assumptions of mass conversion and its uncertainty (Huss, 2013) only apply for glacier-wide estimates (as flux divergence is compensated at the glacier scale). Splitting the glaciers in pieces during the tiling invalidates that assumption. Depending on the size of glaciers in the eastern Tibetan plateau, authors should justify that the tiling is large enough to compensate for this effect (tile size much larger than individual glaciers in the region). I'm expecting that 0.5°x0.5° should not be a problem. However 0.1°x0.1° might be at the very limit, and either require to add uncertainty during the density conversion or to be dropped entirely.

Thanks for you comments. Regarding your first point, we apologize if our previous explanation caused any misunderstanding. As we stated, we did interpolate the missing elevation difference (dh) to calculate glacier mass balance. However, the original, non-interpolated dh values were retained in the final product and this information is clearly documented in the data structure description (originally in the supplementary material but has been moved to the main data description document following the editor's suggestion). When producing glacier-scale products, the decision to include a glacier in the mass balance calculation is based on the non-interpolated dh values. We have revised the corresponding descriptions to avoid any ambiguity.

Regarding your concern about the gridded data product, we agree with your assessment that the

uncertainty should be estimated at the glacier-wide scale. Calculating uncertainty at each grid cell would not accurately capture the errors associated with glacier change. Therefore, we first spatially aggregated the dh values (for valid glaciers), corresponding dh errors, and dh years for each individual glacier. For a specific grid (e.g., 0.5°), the median values of dh, dh error, and dh year were used to represent the grid cell. Then, we calculated the NMAD and the number of pixels representing glacier change within each grid cell to estimate a resampling error. Finally, the mass balance uncertainty for each specific grid cell was calculated based on the dh error, resampling error, area error, and density error.

We also appreciate your comment regarding density errors in small tiles, which may encompass, for instance, the accumulation areas of multiple glaciers without corresponding ablation areas, making it exceptionally challenging to assess density changes accurately. At this stage, it is difficult to rigorously evaluate this uncertainty; therefore, we have removed the 0.1° product as per your suggestion.

3/ Unclear error propagation for regional scale
In the abstract and in line 348: "The average elevation difference for the period from 1970s to 2000 was -9.52 ± 4.16 m, corresponding to a mass balance of -0.30 ± 0.12 m w.e. yr-1".
This is unclear: Is this an estimate for the regional-scale glacier mass change with error propagation, or a per-glacier mean value (so area-weighted with this unit) for both the mean estimate and the uncertainty estimate?
While the mean of these two variables is the same (area-weighted), their uncertainty represent two very different things: that of the total regional mass loss, or that of a single-glacier mass loss on average. If the authors' estimates are intended to represent total regional mass loss, they will need to add additional error propagation steps from the glacier scale to the regional scale, based on correlated errors in density conversion, glacier areas and elevation change. Typically, density conversion errors are assumed 100% correlated, while area not necessarily, and the authors have already estimated the long-range correlations in elevation change errors.

Thanks for your comments. Our results present a per-glacier mean value. To estimate the uncertainty, we utilized the spatial_error_propagation function in xdem to calculate the overall uncertainty of elevation change for all glaciers within each group. Subsequently, we computed the mean of the uncertainties across 261 groups to obtain the total dh error. This calculation incorporates the long-range correlation error within each group.

Following your suggestions, we re-applied the spatial_error_propagation function, utilizing the same variogram of dh error to compute propagated uncertainties for the total region, individual basins, and gridded tiles. During the calculation, we passed an outline of all glaciers within each group to obtain accurate estimates, as recommended by xDEM. However, for groups covering extensive areas, this approach occasionally resulted in memory errors (my workstation has a 128 GB RAM). In such cases, we substituted the glacier outline with a float area value of glaciers to estimate the uncertainty. Out of 261 total groups, 203 were evaluated using the outline of all glaciers. For uncertainties associated with density or area, a 100% correlation was assumed at the regional

scale, and the mean errors in density and area were applied. Finally, we recalculated Equation (8) to determine the mass balance error at the regional scale.

The error statistics at the basin scale (Tables 2 and S3), the group-scale error statistics (Figure 3), and the total regional dh and mass balance uncertainties were all recalculated. The corresponding values in manuscript have been updated accordingly, and these revisions can be reviewed in the tracked document.

4/ Polishing the text
The text would benefit from more streamlining (typos, some unfinished or convoluted statements), and in certain cases some additional English proof-reading.
For instance:
78: "The elevation difference derived from the comparison": "Elevation differences derived from…",
106: "5 to 8 m": typo space missing,
185-187: Gardelle et al. (2013) is the wrong reference for this correction, should be Gardelle et al. (2012),
320: "Statistical uncertainties": no need for "statistical" before "uncertainties" across the entire section,
511: "sources": missing something here… "different"?
These are only examples I took note of here and there, I leave the detailed polishing to the authors.
Thank you for pointing out these issues. we have revised the issues you identified and conducted a comprehensive review of the manuscript. During this process, we corrected typographical errors and refined several incomplete or overly complex statements to enhance the overall clarity and coherence of the text.

Minor comments:
24: "However, the glaciers have been receding during the last decades and are projected to further decline which will profoundly impact the water availability of these larger river systems." This is an over-statement, glacier have moderate impact of water availability, which is only relevant in times of drought. See Gascoin (2023). Should also adjust for this in the introduction.
Thanks for your correction. We revised the sentence to" However, the glaciers have been receding during the last decades and are projected to further decline which will partly and temporarily impact the water availability during drought periods, especially in headwater catchments of these larger river systems." Additionally, we included a related statement in the introduction and cited Gascoin (2023).

Fig. 7/8: These figures related to the analysis of section 2.5.1 look like they were generated with the Python package xDEM, which should then be cited either in the Methods or Code availability, or both.
Thanks. We have incorporated the appropriate citations.

On that topic: It would be great for the authors to share their code through an open repository in a "Code availability" section, for reproducibility of their dataset, and for other researchers to build on

their efforts! (which seems especially important for an ESSD publication)

Thanks. We have included a Code Availability section and shared the processing code on GitHub. See https://github.com/TreeYu123/MB_ETPR-essd-paper-code .

Additional references (not listed in the study)

Gascoin, S. (2023). A call for an accurate presentation of glaciers as water resources. WIREs. Water. https://doi.org/10.1002/wat2.1705

Gardelle, J., Berthier, E., & Arnaud, Y. (2012). Impact of resolution and radar penetration on glacier elevation changes computed from DEM differencing. Journal of Glaciology, 58(208), 419–422.

**Response to Reviewer #2 (Dr. Niccolò Dematteis)**

please consider the following comments and adjust your manuscript accordingly.

i) Table 1 basic information -> try using something less generic than "basic", like general information

Thanks. We have changed the statement.

ii) Table 1 RGI7.0 has been released few years ago . You could use this data to calculate the glacial extent e or at least specify the year of reference of RGI6.0

Since the boundaries of most glaciers in RGI 7.0 are still based on RGI 6.0, and our analysis in the main text also uses RGI 6.0, we have included the acquisition and reference years of RGI 6.0 for the study area. See line?

iii) Figure 2 You should add some details to the caption. For instance, what are the coloured box? Plus, you should use the common convention of flowchart. E.g., a diamond represents a decision, while rectangles are processes. You can refer to the Wikipedia web page https://en.wikipedia.org/wiki/Flowchart or other sources to see what shapes represent what.

Thank you for your suggestions. We have revised the flowchart accordingly: rectangles are used to represent processes, diamonds indicate data inputs, outputs, or the results of a specific process, and multi-documents are used to represent input and output datasets. Additionally, the meanings of some line frames in the diagram have been clarified in the caption. See revised Figure 2.

iv) Figured 3 and 5 you should specify the source of the basemap. If the figures have been created with specific libraries, you should specify this as well.

Thank you for your suggestion. We use Google Terrain Maps as the basemap, and this has been explicitly indicated in the figure caption.

v) Suppl Table 1 Add details to the caption

Thank you for your suggestion. We have revised the title to "The NMAD and Median of elevation difference before (original) and after (co-registration) co-registration processes in off-glacier areas of ETPR"

vi) Suppl Figure 2 A legend is missing (what are the different colours?). Add a background image or country boundaries, because it is very hard to understand the geographical setting.

We have redrawn Figure S2, incorporating the corresponding base map, annotations, scale bar, and other essential elements.

Dataset

Please add the references to the citations in the readme file and convert it in pdf

We have added the corresponding ESSDD citation for the data and contacted the data repository administrator. Once the article is accepted and a new DOI is available, we will update it accordingly. Additionally, we have updated the README file in PDF version. Please check: https://doi.org/10.11888/Cryos.tpdc.301236.

---

## Author Response (AR3)

The authors have satisfactorily addressed all of the comments of my previous review.
I congratulate them on all the work they did to finalize this study! Even if it was a long process since the first submission, the result is very well worth it. I hope this dataset will be useful to many users, and I'm sure it will be durable thanks to their detailed error analysis.

I only noted some minor edits to adjust in the text before publication, see below:
(lines refer to the PDF with track changes)
L27: "temporal specific" → "period-specific"?
L28: "applied DEMs" → "used DEMs"?
L32: "gridded formats at resolutions": change to singular "gridded format at resolution"
L60: and snow melt?
L62: Add Pritchard (2019)?
L73: "Recent research" singular
Spell-out acronyms on first usage in the data section: SRTM, KH-9, ICESat-2, etc
L159: "to estimate ice sheet mass changes"
L192: "could be adjusted" → were they? if yes, need to be clearer!
L212: "hardly possible"?
L260: Add citation to Dowd (1984)
L292-294: It's probably too much detail to cite the name of the function in the package here, could be shortened into "… using xDEM (citation)". Given that xDEM is already cited in the code availability at the bottom, this sentence could also be removed altogether.
L457: Here you could mention that your estimations of long-range spatial correlations for KH-9 DEMs are in line with those of Dehecq et al. (2020).
Fig. 6: Specify that "before" and "after" relate to the coregistration in the caption.
L515 and 518: "Nonetheless/Nevertheless" used twice a row
Additional references
Pritchard, H. D. (2019). Asia's shrinking glaciers protect large populations from drought stress. Nature, 569(7758), 649–654. https://doi.org/10.1038/s41586-019-1240-1
Dowd, P. A. (1984). The Variogram and Kriging: Robust and Resistant Estimators. In G. Verly, M. David, A. G. Journel, & A. Marechal (Eds.), Geostatistics for Natural Resources Characterization (pp. 91–106). Springer Netherlands. https://doi.org/10.1007/978-94-009-3699-7_6
Dehecq, A., Gardner, A. S., Alexandrov, O., McMichael, S., Hugonnet, R., Shean, D., & Marty, M. (2020). Automated Processing of Declassified KH-9 Hexagon Satellite Images for Global Elevation Change Analysis Since the 1970s. Frontiers of Earth Science, 8, 516. https://doi.org/10.3389/feart.2020.566802

**Responses to reviewer #1**
Thank you for pointing out all the formatting issues—we have carefully addressed each one. We have also added the necessary citations for the referenced literature.
Regarding your comment on L192 needing to be clearer, we were actually referring to the discussion in Gardelle (2012) but had missed the citation. We have now added it accordingly. We truly appreciate your thorough review and valuable suggestions!